# Characterization of Palladium Nanoparticles Produced by Healthy and Microwave-Injured Cells of *Desulfovibrio desulfuricans* and *Escherichia coli*

**DOI:** 10.3390/nano9060857

**Published:** 2019-06-05

**Authors:** Jaime Gomez-Bolivar, Iryna P. Mikheenko, Lynne E. Macaskie, Mohamed L. Merroun

**Affiliations:** 1Department of Microbiology, Faculty of Sciences, University of Granada, Campus Fuentenueva, 18071 Granada, Spain; merroun@ugr.es; 2School of Biosciences, University of Birmingham, Edgbaston, Birmingham B15 2TT, UK; I.Mikheenko@bham.ac.uk (I.P.M.); L.E.Macaskie@bham.ac.uk (L.E.M.)

**Keywords:** palladium nanoparticles, microwave injured cells, microwave energy, *Escherichia coli*, *Desulfovibrio desulfuricans*

## Abstract

Numerous studies have focused on the bacterial synthesis of palladium nanoparticles (bio-Pd NPs), via uptake of Pd (II) ions and their enzymatically-mediated reduction to Pd (0). Cells of *Desulfovibrio desulfuricans* (obligate anaerobe) and *Escherichia coli* (facultative anaerobe, grown anaerobically) were exposed to low-dose radiofrequency (RF) radiation(microwave (MW) energy) and the biosynthesized Pd NPs were compared. Resting cells were exposed to microwave energy before Pd (II)-challenge. MW-injured Pd (II)-treated cells (and non MW-treated controls) were contacted with H_2_ to promote Pd(II) reduction. By using scanning transmission electron microscopy (STEM) associated with a high-angle annular dark field (HAADF) detector and energy dispersive X-ray (EDX) spectrometry, the respective Pd NPs were compared with respect to their mean sizes, size distribution, location, composition, and structure. Differences were observed following MWinjury prior to Pd(II) exposure versus uninjured controls. With *D. desulfuricans* the bio-Pd NPs formed post-injury showed two NP populations with different sizes and morphologies. The first, mainly periplasmically-located, showed polycrystalline Pd nano-branches with different crystal orientations and sizes ranging between 20 and 30 nm. The second NPpopulation, mainly located intracellularly, comprised single crystals with sizes between 1 and 5 nm. Bio-Pd NPs were produced mainly intracellularly by injured cells of *E. coli* and comprised single crystals with a size distribution between 1 and 3 nm. The polydispersity index was reduced in the bio-Pd made by injured cells of *E. coli* and *D. desulfuricans* to 32% and 39%, respectively, of the values of uninjured controls, indicating an increase in NP homogeneity of 30–40% as a result of the prior MWinjury. The observations are discussed with respect to the different locations of Pd(II)-reducing hydrogenases in the two organisms and with respect to potential implications for the catalytic activity of the produced NPs following injury-associated altered NP patterning.

## 1. Introduction

Platinum group metals (PGMs) (e.g., Pd, Pt, Ru, Rh, Os, Ir) are widely used as catalysts in many different reactions to obtain valuable products with industrial applications [1]. They are of particular importance due to their unique properties (i.e., high catalytic activity, oxidation resistant properties, mechanical strength, and outstanding resistance to corrosion) [2]. PGM catalysts are used to control the emission of gaseous pollutants from automobiles. Due to this high global demand the price of PGMs has increased substantially since the mid-2000s [3], while high demand for PGMs [4,5] has also increased the focus on recovery processes. Chemical methods offer an alternative for PGM recovery from wastes; these methods include ion exchange, solvent extraction or electrochemical recovery, but they have the challenge of using strong chemicals which are often toxic and environmentally damaging [6]. 

Bacterially-mediated recovery of PGMs is considered as an emerging green and cheap alternative to traditional physical and chemical approaches. Bio-derived methods can exhibit numerous advantages since bacterial species used as templates are easy to grow in large amounts and are capable of rapid metal reduction to form metallic nanoparticles that are comparably active to commercial catalysts in chemical synthesis [7].

Bacteria can interact with soluble metal species in many different ways (e.g., via enzymatic reduction, biosorption, biomineralization, etc.) [8,9,10]. Bacterially-mediated reduction of metals into a neo-catalyst has attracted much interest with other potential applications in, for example, fuel cells [11], decontamination of groundwater [12], and catalytic upgrading of heavy fossil and pyrolysis oils [13,14]. Some microorganisms are able to recover Pd (II) from acidic solutions similar to the conditions that are present in industrial wastes (and from actual waste leachates) and convert waste PGMs into a green neo-catalyst [15,16]. A life cycleanalysis of the latter, as applied to catalytic upgrading of heavy fossil oil, showed theeconomicpotential of this approach even before factoringin the energy (carbon) savings in refinery and mitigation of the high carbon impact and environmentaldamage involved in mining and metal extraction from primary ores [17].

The use of bacteria for synthesis of metallic nanoparticles (NPs) offers the advantage of NPsize control via bio-patterning and the use of enzymes for the Pd reduction avoids the use of toxic chemicals as capping agents that would add to the process cost [18]. Additionally, living systems operate at ambient temperatures, making the process of synthesis of NPs economically attractive. For example, *Desulfovibriodesulfuricans*, a Gram-negative strain, has been shown to use periplasmic hydrogenases supplied with hydrogen to form Pd NPs in the periplasm [9]. NP-synthetic capability has been shown also in other Gram-negative bacteria like *Shewanellaoneidensis*, *Escherichia coli*, and *Pseudomonas putida* [7,19,20,21] as well as Gram-positive bacteria such as *Bacillus sphaericus* and *Arthrobacter oxydans* [22,23]. With the use of modern microscopes, recent studies reported the accumulation of small intracellular Pd NPs in both bacterial types [24], as well as in cell surface layers.Although the former brings possible limitations of substrate access, the use of acetone-washed cells permeabilizes them, whereas NPs stripped of their biochemical scaffold agglomerated and lost activity [25], while partial cleaning altered the catalytic activity as the metal surface wasprogressively unmasked [26]. However, such processingwould add to the production cost andhence this study reports the use of a supported Pdcatalyst made on whole cells.

In addition to cellular location, particle size, and shape, dispersity can play an important role in catalyst reactivity in some reactions [27]. In the case of microbial synthesis of Pd NPs, some studies have shown an influential role of the biological component in the control of shape, size, and distribution of NPs and, as a consequence, their catalytic activity [7]. A possible association of the initial Pd “seeds” with intracellular phosphate structures has been postulated in cells of *Bacillusbenzoevorans*, preventing the Pd NPs from agglomeration [24]. Electron donors such as formate or hydrogen used in NP fabrication can influence the sizes of the biochemically-formed PdNPs and, with this, their electrocatalytic activity [28]. Taking into account these different factors a method of manipulating the formation of NPs to influence their size and distribution could result in a tailored catalyst for increased reaction rates and selectivity in a given reaction.

The main challenges that face the synthesis of nanoparticles are: control of the size and shape of the NPs and monodispersity. It is known that thermal factors can affect the size and uniformity of nanoparticles [29]. Localized heating can be achieved by the use of microwave radiation. Any material (but particularly water) can absorb microwave energy and this is expressed by its dielectric loss factor combined with the dielectric constant. When the microwave heats the desired material through the dielectric loss, it converts the radiation energy into thermal energy [30]. In organic synthesis this has been shown to accelerate processes involved in homogeneous catalysis [31]. The efficiency of microwave energy for the synthesis of a variety of nanomaterials including metals, metal oxides, and bimetallic alloys has been shown [32]. The effect of microwave (MW) radiation on microorganisms has also been studied [33,34]. Some authors noted that application of radiofrequency (RF) microwave radiation (2.45 GHz) altered the activities of some enzymes expressed in *Staphylococcus aureus*resulting from some changes in the cell that could not be explained by the thermal effect [35]. More recently, Shamis et al. [34] confirmed that MW radiation on cells could result in toxic effects when the heating effect was discounted. By modulating the frequency of the MW radiation [36] different biological effects in terms of protein structures were observed, together with alterations in the routes of some biochemical reactions.

In this study MW energy was applied to cells of *E. coli* and *D. desulfuricans* before their exposure to palladium solution. Following the MW treatment, synthesis of Pd NPs was performed using molecular hydrogen as the electron donor. Characterization of size, shape, cellular localization, and atomic structure of the fabricated NPs was conducted by means of scanning transmission electron microscopy (STEM) associated with a high-angle annular dark field (HAADF) detector and energy dispersive X-ray micro analysis (EDX). The use of X-ray diffraction analysis of bulk material was largely precluded by the small nanoparticle sizes and hence poorly resolved powder patterns of the largely amorphous biomaterial [37]. The possible application of the MW treatment to moderate the synthesis of more dispersed and homogeneous Pd NPs is discussedwith referenceto data obtained from high-resolution electron microscopy in conjunction with image analysis.

## 2. Materials and Methods 

### 2.1. Bacterial Strains and Culture Conditions

Two Gram-negative bacterial strains were used in this study, a facultatively anaerobic strain *Desulfovibrio desulfuricans* NCIMB 8307 and the facultatively anaerobic *Escherichia coli* MC4100 as described previously [19,24]. *D. desulfuricans* was grown anaerobically under oxygen-free nitrogen (OFN) in Postgate’s medium C (Sigma-Aldrich) (pH 7.5 ± 0.2) at 30 °C (inoculated from a 24 h pre-culture, 10%*v/v*) in sealed anaerobic bottles [24], while *E. coli* was grown anaerobically (37 °C) on nutrient broth N° 2 (Sigma-Aldrich) supplemented with 0.5%(*v/v*) glycerol (Sigma-Aldrich) and 0.4% (*w/v*) fumarate (Sigma-Aldrich) as described previously [19]. Cells were grown to mid-exponential phase and harvested (Beckman Coulter Avati J-25 Centrifuge, U.S.A) by centrifugation (12,000× *g*, 15 min), washed 3 times in 20 mM MOPS-NaOH buffer (pH 7), concentrated in a small volume of buffer to between 20 and 30 mg dry weight per ml and stored under OFN at 4 °C until next day [38]. Cell dry weight was calculated from optical density (OD_600_) by a previously-determined dry weight conversion factor (mg dry cells= CF × OD_600_ × n, (where n is the dilution factor)).

### 2.2. Microwave Irradiation of E. coli and D. desulfuricans Cells

#### 2.2.1. Microwave Irradiation Conditions

This study was carried out using a portable commercial apparatus (CEM corporation, North Caroline, United States) (CEM Discover SP microwave digestion system; single-mode energy source; 300 W magnetron; ~3 GHz, 300 W). Vials containing cells in 6mL volume re-suspended in 20 mM buffer with concentration between 20 and 30 mg dry weight/ml were exposed in short bursts (10 s) interspersed with periods of 30 s of cooling in ice cold acetone after exposure. This process was repeated three times (total irradiation period of 30 s). During the microwave irradiation sample vials were cooled in hexane.

#### 2.2.2. Microwave Irradiation of Resting Cells Suspended in MOPS Buffer

A 5 mL volume of concentrated cell suspension between 20 and 30 mg/mL in 20 mM MOPS-NaOH buffer (Sigma-Aldrich) pH 7 was added into a 6 mL sealed tube under OFN and treated as above. After microwave exposure, a known amount of treated cells was taken and added immediately to a new sealed tube containing Pd (II) solution (below), representing a final 5 wt% of Pd on the cells. As a control, a 6mL sealed tube under OFN of Pd(II) solution in buffer was added and exposed to microwave radiation under the same conditions as above.

### 2.3. Preparation of Palladium-Challenged Cells

#### 2.3.1. Palladium Solution

For “palladization” of cells a Pd(II) solution was used: 2 Mm Na_2_PdCl_4_ (Sigma-Aldrich, St. Louis, Missouri, United States) pH 2 in 0.01 M HNO_3_ placed in sealed tubes (final volume of 6 mL) and degassed with oxygen-free nitrogen (OFN) under vacuum prior to addition of bacteria.

#### 2.3.2. Formation of PdNanoparticles by Control and MW-Treated Cells

Following microwave treatment, tubes with cells (and untreated controls) were allowed to stand in a water bath (30 min, 30 °C) for uptake of the Pd (II) ions in order to form nucleation sites on the biomass. Hydrogen was added as an electron donor by bubbling H_2_ gas through the suspensions in the sealed bottles (15 min) which were left under H_2_ (24 h) for complete reduction of palladium on the cells (confirmed by assay of residual soluble Pd (II)). Palladized cells were harvested by centrifugation (12,000× *g*, 15 min) and washed with distilled water twice prior to fixation (2.5% glutaraldehyde in 0.1 M cacodylate buffer (pH 7)) for examination by electron microscopy. Controls of palladium-challenged cells without MW treatment were prepared in the same way.

#### 2.3.3. Residual Pd(II) Quantification Using the Tin(II) Chloride Method

In order to confirm complete depletion of Pd (II) ions from the solution, the spectrophotometric tin (II) chloride-based method was used as described previously [39].

### 2.4. High-Resolution Scanning Transmission Electron Microscopy (STEM) with HAADF (High-Angle Annular Dark Field) Detector and EDX Analysis

For STEM analysis, thin sections of palladized MW-treated and non-treated *E. coli* and *D. desulfuricans* cells were prepared according to the method described by Merroun et al. [40]. To determine the location of palladium NPs in the cells, palladized cells were examined under a high-angle annular dark field scanning transmission electron microscope (HAADF–STEM) FEI TITAN G2 80–300 at 300 KeV. For elemental analysis, EDX (energy dispersive X-ray microanalysis of specimen microareas) was used with a spot size of 4 Å and a live counting time of 50s coupled with a high-resolution STEM and HAADF detector. Element co-localizations (Pd, P, S) were enumerated by use of the Manders overlap coefficient (MOC) [41] implemented in ImageJ via the JACoP [38] and co-localization was assumed if the Manders coefficient was greater than 0.9.

### 2.5. Image Processing, Lattice Spacing Determination and Particle Size Analysis 

The HAADF–STEM images were used to determine the average size of Pd NPs produced in different experiments as well as their distribution by means of the image processing software ImageJ (National Institutes of Health, Maryland, United States) [42]. In order to distinguish the Pd nanoparticles on/in cells from background signals and artifacts the same methodology as described by Omajali et al. [24] was used and mean particle size was calculated (mean ± SEM from at least 3 different areas of samples; total NPs analyzed was > 100). Significant differences were assigned using the two sample test of the variance at P = 0.95. The polydispersity index or coefficient of variation was calculated from the particle size distribution dividing the standard deviation of the means by the means [43]. At least 100 particles were counted in each case using ImageJ software. The particle size distribution was estimated using Origin Pro 8. The lattice spacing was determined using ImageJ through profiling of HR-TEM images and compared against lattice spacing of bulk palladium from the database generated using Powder cell 2.400 software (2.4, Informer Technologies, Inc.).

## 3. Results

### 3.1. Microwave-Injury of Bacteria

Examination of the cells by electron microscopy post-injury showed cellular damage as compared to uninjured controls (Appendix A), similar to the response observed by Shamis et al. [34], i.e., shrinkage of the cytoplasmic compartment away from the wall layers with an enlarged periplasmic space. Shamis et al. [34] attributed this to the electromagnetic radiation and not the heating effect. Even though cooling was applied, it was not possible to make this distinction unequivocally using the commercial equipment in this study. Instead, a prior study [44] was carried out using purpose-built equipment that decoupled the electromagnetic and thermal effects at a comparable applied dose (Appendix A). This showed an identical cellular response visually to that using the commercial equipment (with cooling) in the present study and hence, as noted by Shamis et al. [34] the reported cellular response can be attributed to the effect of the MW irradiation.

### 3.2. Examination of the Pd Nanoparticles Produced by Native and MW-Injured Cells of *E. Coli* MC4100 and D. Desulfuricans NCIMB 8307

Controlsof Pd(II) solution in buffer alone exposed to microwave radiation showed no removal of Pd (II) from the solution after microwave exposure usingthe tin (II) chloride method, nor the appearance of any black Pd(0), indicating that microwave irradiation had no active role in the reduction of Pd (II) under the condition tested in this work.

In order to examine and identify the Pd NPs made by native and injured cells, high-resolution HAADF–STEM with EDX was used. For both strains the Pd loading was 5 wt% (1:19 mass of Pd:dry weight of cells). Electron opaque NPs, identified as Pd by EDX (Appendix A) were found in the cell surface layers and within the intracellular matrices (Figure 1B,E and Figure 2B,E). In the case of untreated cells of *E. coli* (Figure 1A–C) large clusters were observed within the intracellular matrices (Figure 1C inset bottom left) and membrane (Figure 1C, main panel, arrowed) at high magnification while MW-treated cells showed apparently more dispersed intracellular NPs with few clusters (Figure 1F).

In contrast, untreated cells of *D. desulfuricans* showed a deposition of surface bound NPs in clusters (Figure 2B,C) in agreement with Omajali et al. [24]. Pd NPs located at the level of the periplasm showed inclusions in the form of nano-branches with sizes ranging from 20 to 30 nm (Figure 2C arrowed), while intracellular NPs were visible which were smaller with sizes between 1 and 5 nm (Figure 2C inset top left). Following MW treatment, and in contrast to *E. coli* (above), the cytoplasmic compartment of MW-treated *D. desulfuricans* remained contracted to reveal NPs in the outer and inner membranes (Figure 2D) with NP-deposition also in the periplasmic space (Figure 2D circled area). The main differences were in morphology of the NPs observed at the level of the surface (Figure 2F) in comparison with untreated cells (Figure 2C), where larger clusters were observed at high magnification. No major differences in number and morphology of Pd NPs were apparent visually by HR-TEM alone within the intracellular matrices in treated (Figure 2C top left) and untreated cells of *D. desulfuricans* (2F bottom right).

The distribution of PdNPs within the intracellular matrices, cell surface layers, and membrane was established by using elemental mapping (Figure 3 and Figure 4). The main elements associated with Pd were phosphorus (P) and sulfur (S). Statistical analysis using ImageJ software [41] was done in order to determine the Manders overlap coefficient to reveal any correlation in localization between Pd and S,and Pd and P in each strain and the effect of the MW treatment on any co-localizations. The Manders overlap coefficient was above 0.9 for both strains and treatments (Figure 3 and Figure 4). According to the statistics analysis done using ImageJ, no differences in the degree of co-location for the selected elements for control and MW-treated cells were observed for either strain (Figure 3 and Figure 4).

The co-location of Pd with S in *D. desulfuricans* is assumed to be PdS resulting from biogenic H_2_S from residual metabolism [24] and the formation of PdS was confirmed in cell surface layers of sulfidogenic bacteria using X-ray photoelectron spectroscopy [45]. In addition, sulfur is present in the amino acids, cysteine and methionine (components of proteins), while phosphorus is ubiquitous within deoxyribonucleic acids, ribonucleic acids, phospholipids, etc., as well as phosphorylated proteins and ATP. The role of these biological components in the patterning of palladium deposition is under current consideration.

### 3.3. Dispersity and Size Distribution of Pd Nanoparticles

Analysis of the nanoparticles size distribution was performed using high-resolution images and ImageJ software [46]. Cells previously MW-treated and then exposed to Pd (II) followed by reduction to Pd (0) were analysed. The mean particle size of the intracellular Pd nanoparticles was 1.28 nm and 1.17 nm for the control and treated cells of *E. coli* MC4100, respectively (Figure 5A,B). 

For *D. desulfuricans* the major differences in terms of shape and size of the NPs were observed on the surface so the analyses were mainly focused on these areas. The corresponding mean particle sizes of membrane Pd NPs produced by treated cells was 1.69 nm and 1.4 nm for control cells (Figure 5C,D, respectively). These differences were significant at P = 0.95 (two sample test of the variance) and hence *E. coli* makes smaller intracellular NPs in response to MW irradiation whereas the NP size increases in the case of *D. desulfuricans* surface-located NPs. However since the differences were small (~10–20%), a mechanistic biological significance cannot be attributed to them at this stage and further work is required to reveal the underlying reasons which may be attributed to the mechanisms of NP deposition in the two strains (see Discussion).

Despite the small differences in mean sizes of the Pd nanoparticles, notable differences were found in their degree of dispersity. The polydispersity value of Pd nanoparticles produced by treated cells of *E. coli* was 0.55 as compared to 0.80 for untreated cells. In the case of *D. desulfuricans* the polydispersity value was 1.26 and 2.07 for MW-treated and untreated cells, respectively. The lower value of polydispersity indexes (32% and 39% lower than controls for *E. coli* and *D. desulfuricans*, respectively) for the two strains resulting from the MW injury suggests a higher degree of homogeneity of the size of nanoparticles. This is accordance with the visual appearance of the cells as noted above.

### 3.4. Crystallinity and Lattice Spacing of Pd Nanoparticles

No differences were observed in terms of crystallinity of the particles under these conditions (Figure 6 and Figure 7). The most representative lattice spacing for *E. coli* was 0.204 nm and 0.213 nm consistent with the (200) facets, and 0.241 nm and 0.23 nm consistent with the (111) facets of Pd, showing a mixed-facet arrangement for both membrane and intracellular Pd nanoparticles (Figure 6A,B).

MW-treated cells of *D. desulfuricans* showed lattice spacing of 0.223 nm and 0.199 nm, again consistent with the (111) and (200) facets (Figure 7A) and, similarly, 0.231 and 0.202 nm for untreated cells. A similar NP structure was observed in previous studies by Omajali et al. [24] of Pd nanoparticles in *D. desulfuricans* with a mix-facet arrangement of (111) and (200) with different crystal orientations in the case of the larger clusters (Figure 7A) when made under H_2_ as in the present study.

## 4. Discussion

This study focuses on the synthesis and characterization of palladium nanoparticles produced by two different bacterial strains that were previously injured via application of microwave energy compared to cells that had no MWexposure. The capacity of these two related genera for the synthesis of Pd NPs is well known via the activity of hydrogenases [9,19] as well as via other (unidentified) mechanisms in cells under conditions in which hydrogenases are not expressed [47]. The chemical reduction of Pd (II) under hydrogen or with formate as an electron donor using killed cells was almost negligible [9,19]. Few studies have been published using cells previously treated with MWenergy and none involve the synthesis of Pd NPs. Previous studies proposed that the application of radiofrequency (RF) energy in *E. coli* under similar conditions to those used in this work might cause electrokinetic modification of the cell surface with a destabilization of the cell membrane [34]. Another observation made by this group was that the application of MW energy resulted in disruption of the cellular membrane and, as a consequence, cytosolic fluids within the *E. coli* cells could pass out through the membrane. However, this was a temporary effect as the shape of the cells was restored within a 10 min recovery period. In the present study the Pd (II) was applied immediately after the removal of the cells from the MW apparatus within 1 min at the early stage of the 10 min “recovery window”. However, follow-up work showed that incorporating the Pd (II) at the outset, during MW exposure, gave similar results as with MW exposure during the pre-Pd (II) period [44] using the experimental setup described here as well as that of the earlier study (Appendix A).

In normal processes of formation of PdNPs on *E. coli*, a possible mechanism to explain the transport of Pd inside the cells was highlighted by Deplanche et al. [7], while a previous report [47] showed that Pd (II) is transported across the membrane through an unknown translocation mechanism. It is known that Ni (II) is a key component in many metalloenzymes [48,49] that are located in the cytoplasm (e.g., ureases and hydrogenases); the latter are also located in the cell membrane and, in the case of *D. desulfuricans*, in the periplasmic space. Deplanche et al. [7] suggested that, due to the similar chemistry of Ni (II)and Pd (II), the latter could be transported inside the cells through the Ni (II) “trafficking system” and deposited as NPs in the cytoplasm as a possible mechanism to forestall cytotoxicity if the Pd (II) is taken up in lieu of Ni (II), but cannot substitute for Ni (II) functionally.

Considering cells exposed to MW energy followed by exposure to Pd (II) solution, apart from the mechanisms mentioned above, additional responses could be activated in the cells that could influence the deposition of Pd inside the cells. According to Shamis et al. [34] cytosolic fluids would be extruded from the cells as a response to the MW radiation and the Pd (II) ions may have more sites becoming available for initial formation of NPs on re-absorption of extruded material along with Pd (II) ions. As a consequence, a higher number of initial binding sites would be occupied by Pd (II) due to the higher accessibility to the binding ligands originating from the cytoplasm caused by the MW. Deplanche et al. [7] confirmed a correlation between the initial uptake of Pd (II) onto cellular components and the initial formation of nuclei at many coordination sites, resulting in smaller Pd NPs per given biomass per constant amount of Pd. Once the effect of MW radiation is finished and cells are recovered in shape and membrane permeability (with re-absorption of the cytosolic fluids [34]) the resulting cells would have a greater proportion of Pd (II) ions contained intracellularly (as compared to surface-localized) than untreated cells. Once the intracellular reduction of Pd (II) into Pd (0) occurs, the association of the resulting Pd NPs with phosphate or sulfide structures would reduce NP mobility, reducing NP agglomeration. The combination of the translocation mechanisms combined with mechanisms activated in the cells as a response to the MW energy may lead to the formation of Pd NPs with higher dispersity than native cells.

A recent study using *S. oneidensis* for the synthesis of Pd/Au nanoparticles showed how a post-treatment consisting of a doping process, carbonization of bacteria, and reduction of graphene oxide avoided NP agglomeration and, as a consequence, increased the dispersity of the nanoparticles, resulting in higher electrochemical activity than a commercial electrocatalyst [50]. Earlier work had shown co-formation of bio-Pd and reduced graphene oxide on cells of *E. coli* [51] but the catalytic activity of the *E. coli* material was not tested. In contrast, for use as a chemical catalyst the cells are not carbonized but are washed in acetone which destabilizes the membrane and lipid structures, making the intracellular Pd NPs more accessible for the reaction. Recent work showed biogenic palladium catalyst of *D. desulfuricans* cells that had been exposed to MW energy had a higher hydrogenation activity and product selectivity in the hydrogenation of 2-pentyne [44], which will be reported as a companion to the present study. Further, follow-up work showed cells of *E. coli* MWinjured under the conditions described here had a similarly increased activity as a selective oxidation catalyst when developed as Pd/Au core-shell NP structures.

By the altered deposition of the NPs observed in the biogenic Pd NPs synthesized by MW-treated cells of *D. desulfuricans* in the present work, a causal relationship may be suggested although conclusive proof is awaited. Since the deposition of the derived Pd NPs of untreated cells differs between *E. coli* and *D. desulfuricans* (Figure 1A and Figure 2A) it may be suggested that both types of cell may have different mechanisms relevant to the synthesis of NPs in terms of shape and location via cellular responses during exposure to MW energy and during the recovery period. The process of the synthesis of Pd NPs in *Desulfovibrio* strains has been studied previously [9,24]. With respect to the derived Pd NPs synthesized by *D. desulfuricans*, with cells exposed to MW energy, the main differences were observed in the periplasm where the hydrogenases are predominantly located [52,53]. A higher porosity of the membrane may facilitate the deposition of the Pd (II) in many different nucleation sites for the initial seeds that without the increased porosity of the membrane caused by the MW energy would not be possible. This hypothesis could explain why the polydispersity index decreases when the cells were previously exposed to MW energy, indicating a higher number of NPs with homogeneous size, given the same dose of Pd (II). This effect was seen regardless of the bacterial strain used.

The relationship between the catalytic activity and the size of the NPs in these two strains has previously been shown [10]. A related study [44] to evaluate the effect of the MW treatment of resting cells (by the method shown in Appendix A) showed the initial rate of conversion of 2-pentyne to be increased by 50% (from 1.1 μmol per litre per second) by Pd NPs made following MW injury of the cells, with selectivity to the desired product cis-2-pentyne, being approximately doubled. One of the major goals in the optimization of the catalytic reactions using NPs as a catalyst is the NPs size control that will help to further understand the relationship between the size and location of the Pd NPs and the product selectivity for a specific reaction. A good example of size control and good dispersion of Au and Pd NPs is the S-layer protein of the Gram-positive strain *Lysinibacillus sphaericus* JG-A12 [54,55]. The monomer of the S-layer offers a good biotemplate for the formation of NPs with a regular structure, pores with identical size (1–5 nm), and good binding sites for Pd(II) such as glutamic and aspartic groups. The crystal structure of the Pd NPs synthesized by the MW-treated cells in the present study did not differ from the untreated cells, showing similar results for *D. desulfuricans* as those obtained by Omajali et al. [24] where twinned NPs were also seen (Figure 7 and Figure 2b). Future studies will focus on comparative studies of the catalytic activity of the Pd NPs synthesized by MW-treated cells versus untreated cells that will inform the controlled synthesis of bio-Pd NPs with higher catalytic activity. 

## 5. Conclusions

This study shows the application of microwave radiation on Gram-negative cells of *D. desulfuricans* and *E. coli* prior to the exposure of Pd (II) solution for the synthesis of PdNPs with a higher degree of dispersity compared to cells that had no MWexposure. The response to the MW on the synthesis of the PdNPs is strain specific. The main differences of the NPs made by treated cells of *E. coli* were at the level of the cytoplasm with an increase of 32% approximately in the level of homogeneity compared to untreated cells. However, the main differences in treated cells of *D. desulfuricans* were observed at the level of the surface with an increase of 39% in the level of homogeneity of the size of the nanoparticles compared to the control.

## Figures and Tables

**Figure 1 nanomaterials-09-00857-f001:**
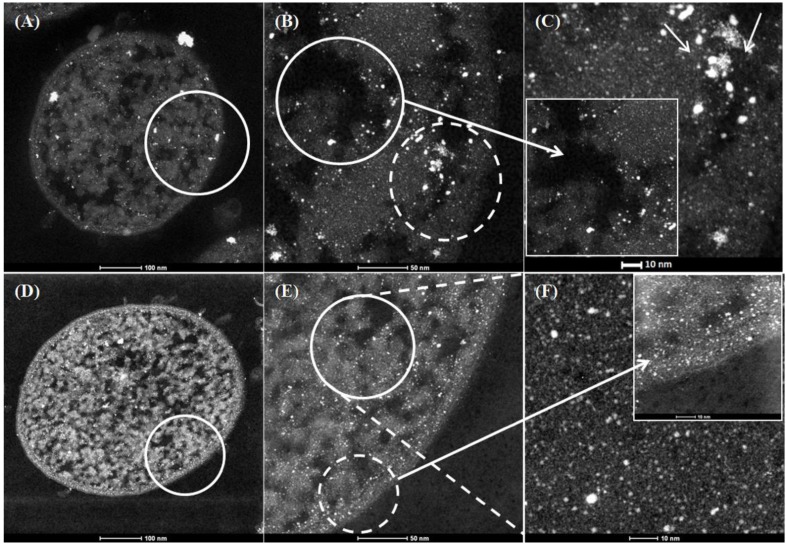
High-angle annular dark field scanning transmission electron microscope (HAADF–STEM) micrographs of Pd nanoparticles synthesized using 5 wt% Pd loading (1:20) on *E. coli* MC4100 from 2 mM Na_2_PdCl_4_ solution, in 0.01 M HNO_3_ using hydrogen as an electron donor without prior microwave (MW) treatment (**A**,**B**,**C**) and with 30 s prior MW treatment (**D**,**E**,**F**).

**Figure 2 nanomaterials-09-00857-f002:**
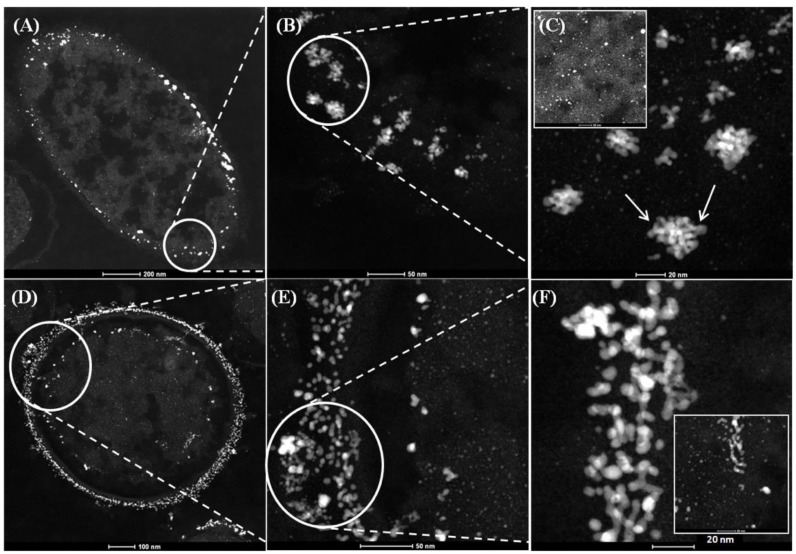
HAADF–STEM micrographs of Pd nanoparticles synthesized using 5 wt% Pd loading (1:20) on *D. desulfuricans* from 2 mM Na_2_PdCl_4_ solution, in 0.01 M HNO_3_ using hydrogen as electron donor without MW treatment (**A**,**B**,**C**) and with 30 second MW treatment (**D**,**E**,**F**).

**Figure 3 nanomaterials-09-00857-f003:**
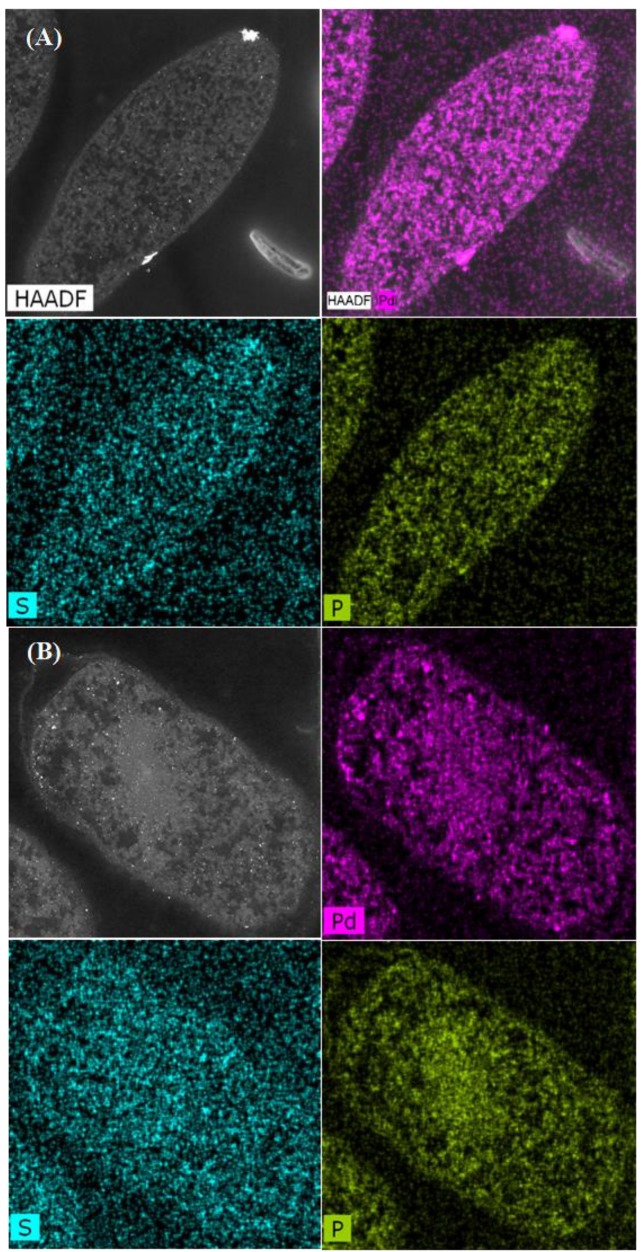
Elemental mapping showing distribution of Pd, P, and S in untreated cells of *E. coli* MC4100 (**A**) and cells treated with MW for 30 sec (**B**). The Manders overlap coefficients were higher than 0.9 for Pd/P and Pd/S in control and MW-treated cells.

**Figure 4 nanomaterials-09-00857-f004:**
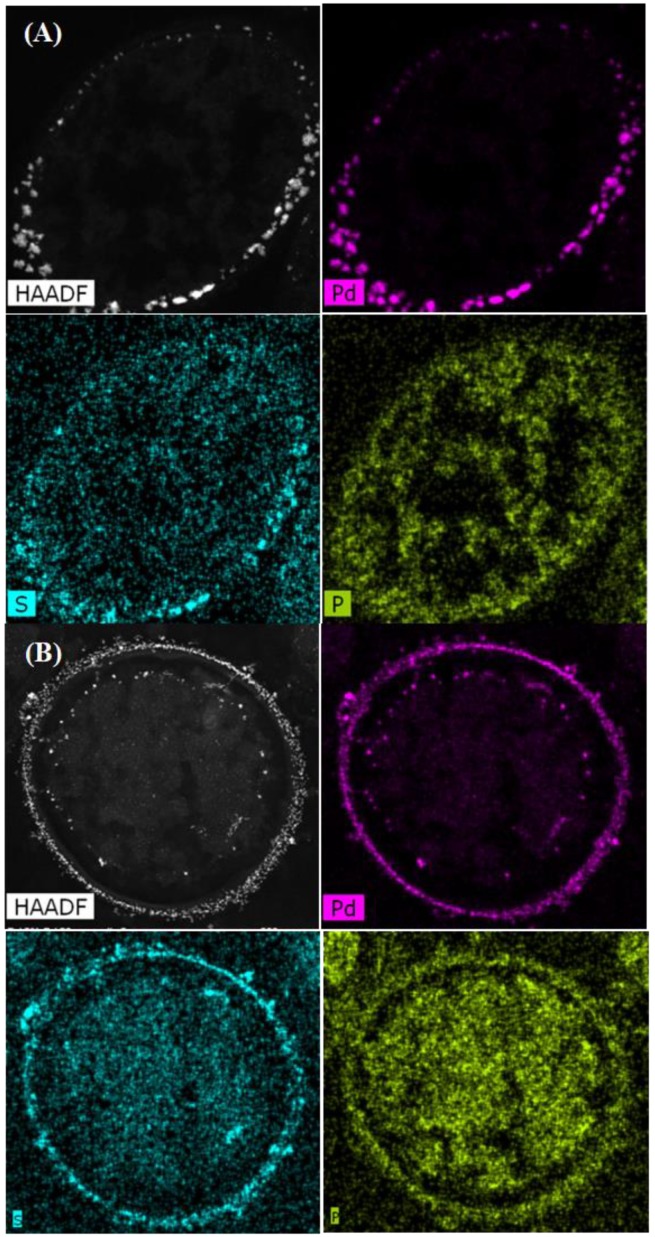
Elemental mapping showing distribution of Pd, P, and S in untreated cells of *D. desulfuricans* (**A**) and cells treated with MW for 30 sec (**B**). The Manders overlap coefficients were higher than 0.9 for Pd/P and Pd/S in control and MW-treated cells.

**Figure 5 nanomaterials-09-00857-f005:**
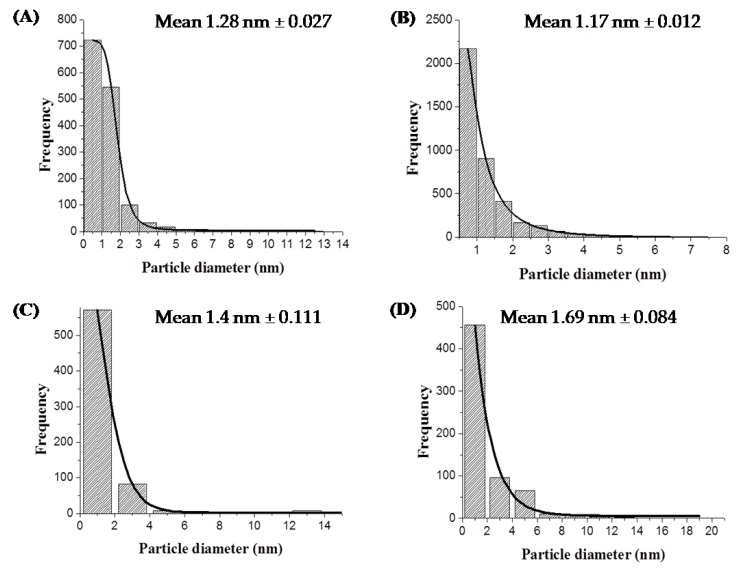
Size distribution and mean nanoparticle (NP) sizes of Pd nanoparticles made by *E. coli* MC4100 untreated (**A**) and MW-treated cells (**B**) and *D. desulfuricans* untreated (**C**) and MW-treated cells (**D**).The mean NP sizes (nm, mean ± SEM) were (**A**): 1.28 ± 0.027; (**B**): 1.17 ± 0.012; (**C**): 1.40 ± 0.11; (**D**): 1.69 ± 0.084.

**Figure 6 nanomaterials-09-00857-f006:**
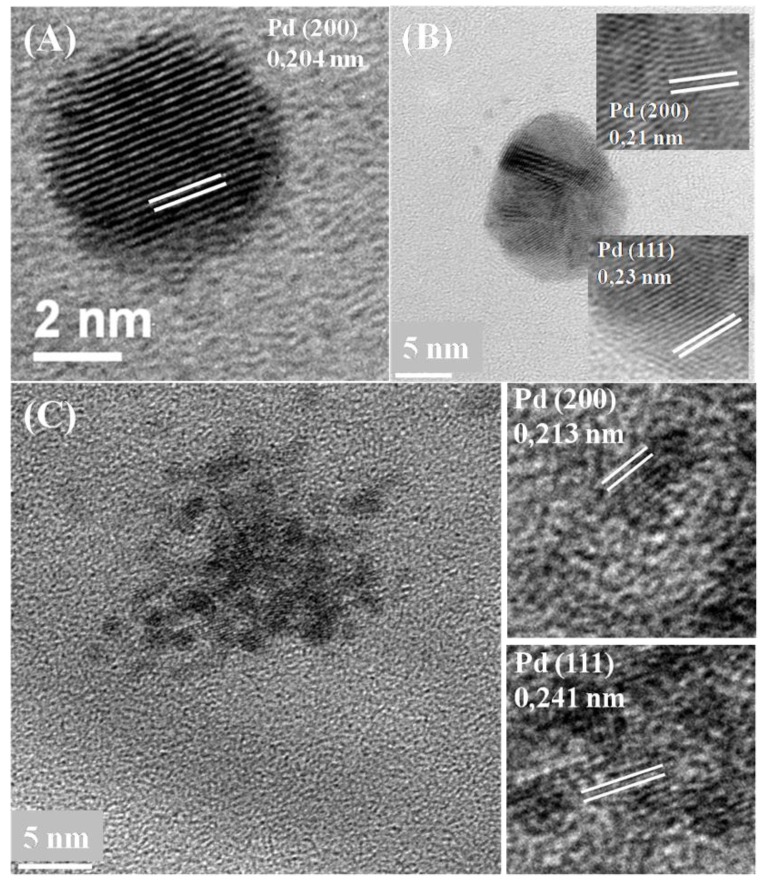
TEM images of intracellular Pd crystals made by *E. coli* MC41005% bio-Pd 30 seconds MW treatment before being exposed to Pd(II) (**A**,**B**) and untreated cells (**C**) revealing lattice spacing in crystals.

**Figure 7 nanomaterials-09-00857-f007:**
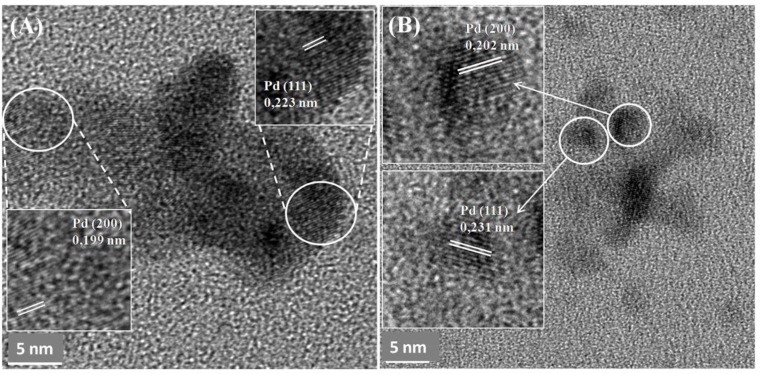
TEM images of membrane-bound Pd crystals made by *D. desulfuricans* 5% bio-Pd 30 seconds MW treatment before being exposed to Pd(II) (**A**) and untreated cells (**B**) revealing lattice spacing in crystals.

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
