# Peer review of "Characterization of Palladium Nanoparticles Produced by Healthy and Microwave-Injured Cells of Desulfovibrio desulfuricans and Escherichia coli"

_nanomaterials, 2019, doi:10.3390/nano9060857_

Reviewer 1 Report

This work is interesting. In general, it is well organized and written. I have no serious comments.

Author Response

No comments need to be added

Reviewer 2 Report

In this study MW energy was applied to cells of E. coli and D. desulfuricans before their exposure  to palladium solution. Following the MW treatment synthesis of Pd NPs was performed using molecular hydrogen as the electron donor. Characterization of size, shape, cellular localization, and atomic structure of the fabricated NPs was conducted by means of Scanning Transmission Electron  Microscopy (STEM) associated with a High-Angle Annular Dark Field (HAADF) detector and energy dispersive X-ray micro anlysis (EDX). The possible application of the treatment to moderate the synthesis of more dispersed and homogeneous Pd NPs is discussed. The manuscript is interesting work, I have some suggestions to further improve the work as below:

·         Scanning Transmission Electron Microscopy is not STEM it is TEM, please fix this.

·         After nanoparticle synthesis how you get rid of the cell debris. It is not better to burn the cells? Please explain and discuss this.

·         Is this method commercially viable as compared to the other synthesis methods? Please explain and discuss in the manuscript

·         Why XRD is not used to analyse the samples

Author Response

We would like to thank the editor and reviewers for contributing to improve the quality of this manuscript. We have addressed all the points made by the reviewers. We hope that all new information included has significantly improved the quality of the revised MS. Our detailed response to the points made by the reviewers is as detailed below. We apologise for various small errors (now corrected) in the submitted version- complying with the submission deadline meant that we had to skip our usual final checks.

Reviewer 3 Report

More details about Residual Pd (II) quantification using the tin (II) chloride method should be provided. Is it possible to extract the obtained particles from bacteria cells and then checked their detail phisicochemical properties as well as their catalytic activity ? beside the fact that authors declared that future studies will focus on comparative studies of the catalytic activity of the Pd NPs synthesized by MW treated cells  I ecauarage the authors to provide some preeliminary data wich will give this manuscript more interesting for readers. 

Author Response

(The authors gave the same response as above.)

Reviewer 4 Report

Synopsis

Gomex-Boliver et al present their work on the effects on the dispersal, size and composition of PdNPs produce by both E. coli and D. desulfuricans in the presence and absences of microwave (MW) pre-treatment using an array of different microscopy techniques. Their work shows there is a change in both the size and morphology for those PdNPs produced by D. desulfuricans, as well as PdNPs, for both treated sets, become more homogenous compared to those untreated with MW.

The paper is well written and easy to follow with a clear story which I commend the authors for as well as that, the images in the manuscript are of high quality.

Overall
The paper shows a number of results un-reported previously and does build on the little work that has gone before. There are however some major points I wish to report which I hope the authors can address to see their work published:
Major Points

Is the change in dispersal/NP size an active/alive process?

The use of ‘sub-lethal’ levels of microwave treatment leaves a few unanswered questions such as:
The evidence to show whether this is a sub–lethal dose or not, i.e. does this have (for example) a 10% fatality rate in both D. desulfuricans and E. coli or are they different?

As stated in the introduction the application of “Microwave energy offers the advantage to be able to heat the sample quickly but not necessarily uniformly, giving localized ‘hot spots’” There is little in the manuscript to cast off this statement in terms of specifications/measurements taken, or replication of the experiments.   
As well as this is the resulting cell death the reason there is a change in NP size/dispersion? Do you get the same result with dead cells? Does Pd in buffer alone result in NP formation?

The use of 5wt% loading of Pd onto cells.
It is unclear of how the authors came to this number without a more detailed methodology or reference to one. Previous publications by this group have shown more in-depth description of this which I’d hope the authors would reference, as such this gives way to another few questions.

Cell mass/number is critical to the formation of nanoparticles given a specific amount of (Pd) ions as previously stated. Also I believe there is an assumption that all the reduced Pd (assayed by the tin (II) chloride method) is deposited on the cell and migrates with the cell upon centrifugation. Could the authors speculate on whether this is the case in both the treated/untreated cells this or provide data to the contrary?

Reference [42]

A large section of the Discussion is dependent on one reference [42] ahead of is publication (described as ‘manuscript in preparation’). As such a large part of the discussion hinges on this data, it would be expected for the potential reviewers to have access to this either shared publically as a preprint or a selection of the data provided for consideration at the time of review and not for publication. As such it is difficult to find the discussion accurate and grounded with peer-reviewed scientific merit.

Supplementary Material

The supplementary material, (Supplementary 1, Supplementary 2) is inadequately referenced and labelled with no apparent legend in the main text to explain which image/graphs refers to which sample/treatment. One of the images (Supplementary 1A) also appears to be out of focus. In the case of Supplementary 2 it is hard to distinguish which peak refers to which element due to the large labels. The inset picture for each spectrum is also unhelpful, is this the area in which the beam was aimed for the measurements?

Minor Points:
General inconsistency of the use of italics for latin names/phrases

Methodology

L119: A reference or equation is required for the calculation of dry weight from OD600.
L134-135: Specifications of “known” volumes
L153-154: Centrifugation needs time/x g,
L158: Spectrophotometric analysis of Pd reduction using tin (II) chloride – The reference [23] just references another paper.   
Figure 5: Use of commas over decimal point.
Results

L199: It is not clear what the authors are referring to with regards to Supplementary 1  
L275: It is not clear how (poly)-dispersity is calculated.

Author Response

We would like to thank the editor and reviewers for contributing to improve the quality of this manuscript. We have addressed all the points made by the reviewers. We hope that all new information included has significantly improved the quality of the revised MS. Our detailed response to the points made by the reviewers is as detailed below. We apologise for various small errors (now corrected) in the submitted version- complying with the submission deadline meant that we had to skip our usual final checks.

Round  2

Reviewer 4 Report

I'm happy with the response from the authors with regards to the revisions and queries I had about the manuscript and I am now satisfied all have been addressed.

I have no further comments or queries to raise with the new manuscript and wish the authors all the best with their publication.